# Kinetic Modeling and Meta-Analysis of the *Bacillus subtilis* SigB Regulon during Spore Germination and Outgrowth

**DOI:** 10.3390/microorganisms9010112

**Published:** 2021-01-05

**Authors:** Jiri Vohradsky, Marek Schwarz, Olga Ramaniuk, Olatz Ruiz-Larrabeiti, Viola Vaňková Hausnerová, Hana Šanderová, Libor Krásný

**Affiliations:** 1Laboratory of Bioinformatics, Institute of Microbiology of the Czech Academy of Sciences, Vídeňská 1083, 14220 Prague, Czech Republic; marek.schwarz@biomed.cas.cz; 2Laboratory of Microbial Genetics and Gene Expression, Institute of Microbiology of the Czech Academy of Sciences, Vídeňská 1083, 14220 Prague, Czech Republic; ramaniuk@biomed.cas.cz (O.R.); olatz.ruiz@biomed.cas.cz (O.R.-L.); viola.hausnerova@gmail.com (V.V.H.); sanderova@biomed.cas.cz (H.Š.); krasny@biomed.cas.cz (L.K.); 3Bacterial Stress Response Research Group, Department of Immunology, Microbiology and Parasitology, University of the Basque Country UPV/EHU, 48940 Leioa, Spain

**Keywords:** *Bacillus subtilis*, SigB, gene regulatory networks, computational modeling, promoter sequence analysis

## Abstract

The exponential increase in the number of conducted studies combined with the development of sequencing methods have led to an enormous accumulation of partially processed experimental data in the past two decades. Here, we present an approach using literature-mined data complemented with gene expression kinetic modeling and promoter sequence analysis. This approach allowed us to identify the regulon of *Bacillus subtilis* sigma factor SigB of RNA polymerase (RNAP) specifically expressed during germination and outgrowth. SigB is critical for the cell’s response to general stress but is also expressed during spore germination and outgrowth, and this specific regulon is not known. This approach allowed us to (i) define a subset of the known SigB regulon controlled by SigB specifically during spore germination and outgrowth, (ii) identify the influence of the promoter sequence binding motif organization on the expression of the SigB-regulated genes, and (iii) suggest additional sigma factors co-controlling other SigB-dependent genes. Experiments then validated promoter sequence characteristics necessary for direct RNAP–SigB binding. In summary, this work documents the potential of computational approaches to unravel new information even for a well-studied system; moreover, the study specifically identifies the subset of the SigB regulon, which is activated during germination and outgrowth.

## 1. Introduction

Transcription and expression of the physiologically relevant genes is essential for adaptation of organisms to changing environmental conditions. Uncovering the nature of gene regulatory networks is one of the core tasks of systems biology. Identifying direct regulons (group of regulated genes) of sigma factors can be considered as a basic element of this task for prokaryotic organisms where sigma factors are subunits of RNA polymerase (RNAP) that are critical for recognition of promoters (DNA sequences where gene transcription starts [1]). A number of tools for gene regulatory network inference were developed in the last 20 years (a comprehensive review of the methods for gene networks inference can be found in Wang et al. [2] and Loskot et al. [3]). Several such tools (ARACNE [4], and network BMA [5]), including our tool CyGenexpi [6], were integrated into the systems biology platform Cytoscape (http://www.cytoscape.org/). Advances and limitations of network inference methods were reviewed by [7], and substantial work on reconstruction of sigma factor-controlled networks was also performed by Tiwari and Chauhan [8,9], as well as in *Bacillus*, particularly by Nannapaneni [10]. However, it has been shown that using only one source of data for network inference (e.g., only chromatin immunoprecipitation sequencing [ChIP-seq], RNA sequencing [RNA-seq], or literature mining) can be misleading. Therefore, combining multiple sources is necessary [11].

*Bacillus subtilis* is a Gram-positive model organism that survives unfavorable conditions as an endospore. Subsequent spore germination and outgrowth are complex processes [12] that require extensive changes in gene expression that involve a number of sigma factors. *B. subtilis* contains one main (primary, housekeeping) sigma factor—SigA, which regulates gene expression mostly in exponential phase [13,14], as well as 18 alternative sigma factors [15,16,17,18] and one sigma-like factor—Xpf [19].

SigB is the general stress response factor, helping the cell resist oxidative stress; moreover, it also protects cells against heat, acid, alkaline, or osmotic stress [20]. RNAP holoenzyme containing SigB recognizes GTTTaa and GGG(A/T)A(A/T) sequences as the −35 and −10 regions (with respect to +1 transcription start). These two sequences are separated typically by 13 to 15 nucleotides [18]. The *sigB* gene itself is transcribed from two promoters, one SigA-dependent and the other SigB-dependent [21]. The activity of the SigB protein is regulated by a partner-switching signaling network that involves anti-sigma, anti-anti-sigma factors, as well as phosphatases that act upstream of the “anti” factors. Under non-stressing conditions, SigB is typically in complex with its anti-sigma factor, RsbW, and inactive. RsbV, the anti-anti-sigma factor, is under these conditions phosphorylated and unable to interact with RsbW. When stress is detected by the cell, phosphatases RsbQP and/or RsbTU dephosphorylate RsbV, which subsequently interacts with RsbW, and SigB is released and activates its target genes, including its own operon, which also includes genes for RsbW and RsbV [22]. However, as experimental data show [23], SigB is expressed also during germination and outgrowth where the stress conditions are not expected.

Currently, according to SubtiWiki (http://www.subtiwiki.uni-goettingen.de/[24]), there are 217 genes known to be in the SigB regulon. Other works identified additional SigB-regulated genes by using various methods mostly in stress-induced systems [18,25,26,27,28]. Combining all the available data sources, the SigB regulon currently consists of 411 genes. It is most likely that not all of these genes are activated during particular conditions.

To advance our understanding of the SigB regulon and the germination-outgrowth process in *B. subtilis*, we applied a combination of computational and experimental approaches. First, we extracted and combined the data from the previous experiments obtained both from literature and the SubtiWiki database. Subsequently, in order to identify genes regulated by SigB, we computationally modelled the gene expression profiles recording the activity of the cell regulatory networks under given experimental conditions (here the germination and outgrowth) using SigB as a regulator, and those satisfying selection criteria were identified as members of the active regulon. Furthermore, for the genes of the SigB regulon reported as alternatively controlled by other sigma factors, we also computed the models of their regulation, and, where the SigB profile alone could not explain the regulation, other regulators (i.e., sigma factors) were suggested. Analysis of the promoter site binding motifs with respect to the above-mentioned analyses allowed us to define more precisely the structure of the sequence, which is necessary for SigB to bind and activate transcription. The results were then validated experimentally. Using the results of the kinetic modeling and promoter sequence binding motifs analysis, we were able to identify a core set of genes controlled by SigB, which is specifically expressed during germination and outgrowth.

## 2. Materials and Methods

### 2.1. Data Acquisition

#### 2.1.1. Time Series of Gene Expression

We downloaded the *B. subtilis* transcriptomic microarray data from GEO (http://www.ncbi.nlm.nih.gov/geo/query/acc.cgi?acc=GSE6865), consisting of 14 time points (0, 5, 10, 15, 20, 25, 30, 40, 50, 60, 70, 80, 90, and 100 min) during germination and outgrowth [23]. The dataset contains time series of expression of 4008 genes. Briefly, spores of *Bacillus subtilis* 168 were generated by growing cells in a defined MOPS medium at 37 °C and shaking for 4 days. Spores were then thermally activated at 70 °C for 30 min in germination medium. The release of dipicolinic acid in the medium during spore germination was monitored using the terbium fluorescence assay. Samples for RNA isolation were drawn at regular intervals during germination and outgrowth. RNA was isolated from spores and outgrowing spores and Cy-labeled cDNA was produced by reverse transcription using Cy-labeled dUTP. Samples were hybridized to microarray slides; microarrays were scanned using an Agilent G2505 scanner. Data from replicates were averaged, and the original log_2_-based data were exponentiated.

#### 2.1.2. SigB Regulon

The SigB regulon was compiled from the published data [18,25,28,29,30] and from the SubtiWiki database (http://www.subtiwiki.uni-goettingen.de/ [31]) The data in SubtiWiki contain a collection of experimentally validated regulatory relations of *B. subtilis* genes constructed by surveying literature references, including interactions found regardless of the experimental conditions under which they were obtained. The literature data are mostly results of binding experiments obtained under stress conditions. All this results in overlaps of different sigma factors and cofactor regulons within the regulon of SigB. Such interactions were also considered in the analysis below.

### 2.2. Kinetic Model of Gene Expression

We used the model originally developed by Vohradsky [32] and further extended in the works of to and Vu [33,34,35]. The model was implemented as a Cytoscape plugin (www.cytoscape.org) as an R package and a command line tool [6]. In this paper, we used the command line version executed from a Matlab script. All further computations were performed in the Matlab environment. The script executables and the Cytoscape plugin can be downloaded from https://github.com/cas-bioinf/genexpi/wiki. The model as implemented in this paper was described in detail in our previous work [36]. Here, we only briefly mention its principle and a model of constant rate of expression used in the data preprocessing.

The relation between the rate of accumulation of the transcribed mRNA and the sigma factor amount can be described mathematically by a sigmoid with parameters reflecting the strength of binding, reaction delay, and mRNA degradation rate. The model used in this study has the following form:(1)$$\frac{dy_{i}}{dt}=k_{1i}\frac{1}{1+\exp[-(\sum {}_{j}w_{ij}R_{j}+b_{i})]}-k_{2i}y_{i}$$
where *y_i_* represents the amount of the target genes mRNA, and *R_j_* is the amount of the *j*-th sigma factor modulated by parameter *w_ij,_* corresponding to the binding strength to the promoter. The *b_i_* corresponds to the reaction delay. Accumulation of the gene’s mRNA is diminished by degradation described by the term *k*_2*i*_*y_i_*. In the data preprocessing step, we also considered a constant rate of expression model where
(2)$$\frac{dy_{i}}{dt}=k_{3i}-k_{4i}y_{i}$$

Here, *y* is the expression of the target gene as a function of time, and *k*_3_ and *k*_4_ are mRNA synthesis and degradation rate constants, respectively. When a gene expression profile is fitted by the constant synthesis model, it means that its synthesis is not affected by amount changes in the sigma factor. Such genes have to be excluded from the analysis, as when using the Equation (1) they can be fitted with any profile and introduce false positive results.

Since the expression data is noisy, we interpolated and smoothed them prior to computation with a Sawitzky–Golay filter. The smoothing achieved more robust results with respect to high-frequency phenomena expected in gene expression, while preserving the characteristic low-frequency phenomena. A further advantage to smoothing is that it let us subsample the fitted curve at arbitrary resolution. We subsampled the profiles at 1 min time steps, which allowed us to integrate Equation (1) accurately using the computationally cheap Euler method. Optimization of the parameters of the model for individual sigma factor-transcribed gene combination was performed using a simulated annealing scheme by minimization of an objective function
(3)$$E=\surd \sum {\left(y-\tilde{y}\right)}^{2}$$
where *y* represents the experimental mRNA’s amount time series and $\tilde{y}$ represents the time series computed using the model Equation (1). Furthermore, a regularization term was added to the objective function to penalize biologically implausible values of the parameters. The regularization is non-zero if either (a) *k*_1*i*_ allows maximal measured transcript level to be achieved in less than 1 min starting from zero, or (b) regulatory response is very steep with |*w_i_* (t) |> 10 for some t, or (c) the regulatory interaction is never saturated with |−*w_i_* (t) + *b_i_*| > 0.5 for all t. For each profile, the optimization was repeated 256 times with random values of initial parameters estimates. From the 256 runs, the parameters giving the smallest E were selected. The goal was to identify parameters that would fit the measured expression profiles of the given regulated gene with the SigB profile as the regulator within the confidence interval. Where such parameters were found, regulatory interaction between a sigma factor and a gene was considered proven.

### 2.3. Data Preprocessing

Prior to any computations, as mentioned in the preceding paragraph, raw gene expression time series were interpolated to 1 min intervals and smoothed using Golay filters (Matlab function smoothdata, method sgolay; Appendix A). Then, several constraints were introduced: (1) time series of gene expression of the genes whose maximum in expression profile was smaller than an arbitrarily chosen value of 300 were excluded, as the low values of expression profiles bear large variance that can lead to misinterpretation of the modeling results. The threshold level was based on the observation that the variance of the microarray quantified expression values rapidly increases with decreasing magnitude of the signal and for low levels of expression can reach values higher than 50% of the mean (defined as coefficient of variation (CV)). (2) The genes that could be modeled with a constant rate of synthesis model (Equation (2)) were excluded, as they could be modeled by any profile and could bring false positive predictions. (3) A random expression profile created by randomization of the SigB expression profile was used as the regulator profile to model the expression profiles that were not excluded in the previous steps. The modeling with the randomized profile was performed 10 times for all genes that were not excluded in the previous step, and those gene expression profiles that were modelled at least once by the random profile were excluded. The purpose was the same as in the previous case-exclusion of the genes that could lead to false positive predictions.

### 2.4. Promoter Binding Motif Analysis

For identification of the SigB binding motifs, we used the information from Appendix A of the Appendix A of the paper of Nicolas et al. [18]. Appendix A contains sequences of the promoter regions containing two motifs (−35 and −10) found to be present for different sigma factors including SigB. The table contains also “upshift” locations (proxy for transcription start site) for all transcription units (operons). Using the upshift locations, we extracted the sequences <−80; +40> nucleotide range from the upshift position. In these sequences, we used two methods for identification of the binding site motifs: exact match and consensus motif search. For the latter, we first identified the consensus motif from the motifs in Appendix A of Nicolas et al. The computed logo is shown in Figure 1. Each motif was converted to meme format, and the fimo utility (meme-suite.org/doc/fimo.html) was used to search the motifs within the extracted sequences (fimo—norc—thresh 0.001).

For the exact match search, we directly used the sequences of the motif as indicated in the promoter sequences of Appendix A of Nicolas et al. The two results were merged and combined with the information about corresponding transcription units.

### 2.5. Transcription In Vitro

#### 2.5.1. SigB Cloning

The *sigB* gene was amplified using *B. subtilis* BaSysBio genomic DNA [18] with the primers 1004/sigB_F 5′-ggaattcCATATGacacaaccatcaaaaac-3′ and 1006/sigB_R_His 5′-ccgCTCGAGcattaactccatcgagggatc-3′. The resulting DNA fragment was cloned into expression vector pET-22b (comprising inducible promoter and 6xHis-tag) using NdeI and XhoI restriction enzymes. The resulting construct was validated by sequencing transformed into *E. coli* BL21 (DE3) competent cells, and the strain was named LK#1207.

#### 2.5.2. Media, Growth Conditions, Protein Purification 

For protein purification, the strains were cultured in Luria–Bertani (LB) medium at 37 °C with continuous shaking. *Bacillus subtilis* RNAP *rpoE* (LK 637, [37]) with a His10-tagged β’ subunit was purified as described previously [38]. SigB was overexpressed from strain LK#1207. The strain was grown in 2L of liquid LB medium at 37 °C, and when the bacterial culture reached OD_600_ 0.6–0.7 (mid-logarithmic phase), it was transferred to room temperature, and SigB overexpression was induced with 0.8 mM isopropyl β-D-1-thiogalactopyranoside (IPTG) for 3 hours under constant shaking (120 rpm). Purification of SigB via 6x His-tag using affinity chromatography was performed as described previously [38]. Delta subunit (RLG7023 [39]) of *B. subtilis* RNAP was purified as described [39].

#### 2.5.3. PCR of DNA Templates

Linear PCR products of putative promoter regions were used as templates for in vitro transcription assays. The primers are listed in Appendix A. Putative promoter sequences were PCR-amplified using wild type (wt) *B. subtilis* gDNA as the template. All PCR reactions were performed using the Expand High Fidelity System (Roche). The purification of PCR constructs was performed using QIAquick Gel Extraction Kit from Qiagen according to the manufacturer’s protocol.

#### 2.5.4. In Vitro Transcription Assays

The *B. subtilis* RNAP core was reconstituted with saturating concentrations of SigB and delta. Reconstitutions were performed in a glycerol storage buffer (50 mM Tris-HCl (pH 8.0), 0.1 M NaCl, 50% glycerol) for 10 min at 37 °C. *E. coli* RNAP holoenzyme was purchased from NEB (cat# M0551S).

Multiple round transcription reactions were carried out in 10 μL reaction volumes with 60 nM *B. subtilis* RNAP holoenzyme and 50 ng of linear DNA template. The transcription buffer contained 40 mM Tris-HCl (pH 8.0), 10 mM MgCl2, 1 mM dithiothreitol (DTT), 0.1 mg/mL bovine serum albumin (BSA), and 150 mM KCl. ATP and GTP were 400 μM, CTP was 200 μM, and UTP was 10 μM plus 2 μM radiolabeled [α-^32^P]UTP.

All transcription experiments were performed at 37 °C. Transcription was induced by adding reconstituted RNAP holoenzyme and allowed to proceed for 15 min. Transcriptions were stopped with equal volumes (10 μL) of formamide stop solution (95% formamide, 20 mM EDTA (pH 8.0)). To generate molecular size RNA marker, we used the same conditions. The only differences were the usage of *E. coli* RNAP and the templates were DNA fragments of the *hcr* gene and plasmid pLK1 [40]. These templates were combined in one reaction and yielded a ladder of 108, 145, 201, 253, 300, 357, and 407 nt. Samples were loaded onto 7 M urea–7% polyacrylamide gels and electrophoresed. The dried gels were scanned with Molecular Imager FX (Bio-Rad) and were visualized and analyzed using the Quantity One software (Bio-Rad).

## 3. Results and Discussion

### 3.1. Modeling of the SigB Regulon

In this paragraph, we discuss the use of the expression profile of SigB to identify genes that could be, from the kinetic point of view, controlled by SigB. This was achieved by modelling of their expression profiles using Equations (1) and (3). Those genes for which the model well fitted their expression profiles were identified as potentially controlled by SigB.

Out of the 411 genes (217 genes from SubtiWiki, an additional 194 compiled from the literature), 260 were reported to be controlled exclusively by SigB (the following results are summarized in Appendix A). For the remaining 151 genes, numerous other regulators and sigma factors were found to participate in the control of their expression, depending on the conditions. According to SubtiWiki and the bibliography, 2 regulators, including SigB, were in the reported SigB regulon found for 88 genes, 3 regulators for 39 genes, 4 regulators for 13 genes, 5 regulators for 7 genes, and 6 regulators for 3 genes. Altogether, 37 regulators including sigma factors were reported to participate in the control of the SigB regulon; the regulon overlap was rather high. Most of the genes reported in the literature as co-controlled by other factors also belonged to the SigA regulon (87 genes) (the competition between SigB and SigA was reported [18]). Other sigma factors are (listed in descending order of number of genes reported to be co-controlled by them) SigM (20), SigG and SigF (16), SigW (14), SigX (10), SigH (6), SigE (5), and SigD and SigI (1). Of the other transcription factors, CcpA, involved in carbon catabolite repression [41] was found 13 times, while the others appeared between 1 to 9 times. We emphasize that these observations were compiled from numerous articles, where the results were obtained under diverse experimental conditions. Our goal then was to identify those regulators (i.e., sigma factors including SigB) that could participate in the control of the SigB regulon under the conditions reflected in the time series analyzed here. We applied the model defined in Equation (1) to the germination and outgrowth of gene expression data from a GEO database (http://www.ncbi.nlm.nih.gov/geo/query/acc.cgi?acc=GSE6865) that contained expression profiles of 4008 genes measured at 14 time points (the experimental conditions are briefly summarized in the Materials and Methods section), the pre-processing excluded 1349 genes with low expression, and 135 genes with constant rate of expression. Of the remaining 2524 genes, another 327 genes with profiles that had been at least once modelled with the random expression profiles were removed (see Materials and Methods, data preprocessing).

For each target gene, all combinations of its reported regulators (single and/or multiple) were modeled and the ability of the model to fit the experimental data was assessed (see Materials and Methods, Section 3.2., Equation (1)). Although the experimental data list some genes with four, five, and even six regulators, we modeled only the combinations with maximally three regulators, as more than three regulators concomitantly acting on one gene at a time is improbable. Furthermore, using more regulators could lead to overfitting, with weights of some regulators close to zero. Selection of the regulators controlling each gene was then performed with respect to the best goodness of fit of the model to the experimental expression profile. Where an equivalent fit quality was achieved for different regulators, or a combination of regulators, we used the following rules: (i) the minimal number of regulators satisfying goodness of fit was selected (e.g., if an equal fit was achieved for SigB only and a combination of SigB plus SigA, then the single SigB regulator was selected); (ii) if SigB was found as one of the regulators, this regulation was preferred; and (iii) if the error of fit was the same for some regulators or a combination of regulators, they were listed as alternatives.

Modeling results are summarized in Appendix A. Of the 411 genes of the documented SigB regulon, 51 were found to be expressed at low levels (see constraints in Materials and Methods Section 3.3), and 123 did not pass the other constraints defined in the Materials and Methods section. For 63, we did not find parameters that could successfully model their expression profiles; these genes were considered to not be controlled by SigB under our conditions. For 25 genes, SigA, instead of SigB, was found as their most probable regulator (*apt*, *atpC*, *csbA*, *hemA*, *hemX*, *nhaC*, *opuD*, *pnpA*, *queA*, *rbfA*, *recO*, *rpsO*, *tgt*, *tmk*, *uvrA*, *uvrB*, *uvrC*, *yhbJ*, *yhcA*, *yhcC*, *yhdH*, *yhgE*, *yocJ*, *yozB*, *yybT*). For 17 genes (*ctc*, *gtaB*, *menC*, *menE*, *nadE*, *yebG*, *yitT*, *ykuT*, *ywsA*, *infB*, *smpB*, *ydaJ*, *ydaK*, *ydaL*, *yebE*, *yoaA*, *ypuB*), the best fit was obtained when SigB and SigA acted together. In summary, the modeling confirmed 148 (36%) genes that were during germination and outgrowth controlled by SigB alone (94) or with a possible participation of other regulators (54).

Finally, we found that of the 46 genes reported as controlled by three or more regulators, 11 were equally well modeled by SigB alone (*gabD*, *katX*, *rsbU*, *V*, *W*, *X*, *yfhE*, *F*, *yxjI*, *yugU*) or by SigB together with one or more reported regulators. No model could be found for six genes (*ylxP*, *Q*, *R*, *S*, *spo0E*, *yqjL*). Generally, in all remaining cases, two regulators were always sufficient to model the target gene expression profile equally well as if more regulators were employed.

### 3.2. Binding Motif Analysis of Predicted SigB-Dependent Transcription of the Genes Expressed during Outgrowth

In order to identify the genes that are actually controlled by SigB during germination and outgrowth, we combined promoter binding site composition (the presence of −35, −10, and their spacing) of the known SigB-dependent genes with their expression kinetics analysis.

We took all genes of the documented SigB regulon (Appendix A) and we searched for the −35 and −10 motifs in the <−80; +40> region (numbering according to a putative transcription start sites, [18]). We evaluated their mutual position and their distance from each other (Appendix A). The same analysis was performed on subsets of genes that we found by the kinetic modelling to be under the control of SigB only (type 1), genes where other regulators besides SigB were found (type 2), and genes where no regulation was identified or that were excluded during preprocessing (type 3). Figure 2 shows the relative representation of the binding motifs and spacer length distributions in these three groups. The distance between the promoter motifs (spacer length) ranged from 1 to 50 bp, where the highest deviation was observed for the excluded (type 3) genes.

In total, 95% of type 1 genes possessed the −35 motif, 78% the −10 motif, and 73% of these genes contained both motifs. Similarly, 97% of type 2 genes had the −35 motif and 66% the −10 motif; 63% contained both motifs. To the contrary, only 53% of type 3 genes had the −10 motif and only 40% of them had both. Importantly, the distance between the binding motifs was more consistent for type 1 and 2 genes where the mean distance was around 15 bp, while for the excluded genes, the mean distance was 18 bp. The largest deviations were observed for the genes for which the kinetic modeling did not find any regulator, or those that were excluded for low expression or flat expression profile (type 3). Therefore, this analysis clearly specified genes whose binding site composition determines that they could be regulated by SigB only and those that require during germination and outgrowth additional/different regulators.

### 3.3. Experimental Verification of Selected Promoters

Finally, we performed in vitro transcriptions in a defined cell-free system for selected genes with different organization of SigB-binding motifs in the promoter region. The genes were selected according to the motif organization and the shape of its gene expression profile (see Figure 3). We used *B. subtilis* RNAP and DNA fragments containing the promoter regions. Altogether, 16 promoter regions were selected. For the full list, see Table 1. These promoters were divided into three classes (class I, II, and III). Class I promoters (10 promoters) contained canonically spaced (by 13–15 bp) −35 and −10 elements. Class II promoters (two promoters) contained only the −10 element, highly resembling the consensus sequence. Class III promoters (four promoters) contained both −35 and −10 elements, but their spacing was more than 14 bp. The genes were also chosen for the shape of their gene expression profile. Most of them were correlated with the shape of the SigB expression profile, with the exception of *pgcA* and *yqhY*, for which the model was fitted with the negative value of the parameter *w*, suggesting inverse correlation with the profile of SigB.

Figure 3 shows transcriptions from eight class I and two class II promoters, in all cases with a clear positive result. However, two class I (cadA, rsgA), one class II (yhfP), and all class III promoters (xynA, yhfI, pfkA, uppS) were inactive. The transcriptional inactivity of class III promoters with RNAP–SigB was anticipated—the −10 elements were not strong enough to act as promoters on their own and the spacer distances to their respective −35 elements were too long. Hence, for genes whose promoters were inactive in vitro, it is apparent that additional transcription factors not included in the in vitro assays are required.

Of the 10 class I genes, the *yqhY* gene was selected for its inversely correlated profile with the expression profile of SigB. The SigB binding motif was present and the promoter was active in vitro (Figure 3). In the *yqhY* promoter region, we also identified a SigA-dependent promoter-like sequence overlapping that one of SigB. However, this promoter was not active in vitro (data not shown). The phenomenon of inversely correlated profiles was also observed for some other genes (see Appendix A) as well reported in a previous study [36]. The exact mechanistic interplay of RNAP holoenzymes/transcription factors at such promoter regions is currently unknown.

Finally, Figure 3D shows SigB-dependent transcription from *PglyA*, a newly found class II promoter not previously known to be SigB-dependent. The *glyA* gene encodes serine hydroxymethyltransferase involved in purine nucleotide metabolism. Its expression is known to be controlled by SigA and the PurR repressor [42].

Taken together, both computational and experimental analyses show that for a gene to be controlled by SigB, it is necessary that the promoter sequence contains both −10 and −35 binding motifs with the spacer length in the range of 15+/−2 nucleotides (with the exception of *glyA*, where the −35 motif was not identified as being significant, but transcription was confirmed). Figure 2 shows substantial differences between the genes that were previously proposed to be under the control of SigB, and those genes that were excluded on the basis of our kinetic modeling. The kinetics of the genes that were excluded were not coherent with the gene expression model, and, importantly, their promoter sequences did not contain full binding motifs with the correct spacing. If we combine all the above mentioned criteria and select the genes that satisfy them, we obtain a core set of 146 genes representing 115 operons that are controlled by SigB during spore germination and outgrowth representing 35% of all genes reported as being the SigB regulon (*aag*, *aldY*, *atpC*, *bmrU*, *clpC*, *copB*, *cpgA*, *csbB*, *csbC*, *csbD*, *csbX*, *csoR*, *ctc*, *ctsR*, *cypC*, *dps*, *galK*, *galT*, *gspA*, *gtaB*, *katE*, *katX*, *malS*, *mcsA*, *mcsB*, *menC*, *mgsR*, *nagBA*, *nhaX*, *ohrB*, *opuD*, *opuE*, *pgcA*, *phoH*, *pth*, *radA*, *rbfA*, *rnr*, *rpe*, *rpmEB*, *rsbRD*, *rsbV*, *rsbW*, *rsbX*, *rsoA*, *sigB*, *tmk*, *truB*, *trxA*, *ung*, *yaaI*, *ybyB*, *ycbP*, *ycdF*, *ycdG*, *yczO*, *ydaC*, *ydaD*, *ydaE*, *ydaG*, *ydaJ*, *ydaK*, *ydaL*, *ydaM*, *ydaN*, *ydaP*, *ydaS*, *ydaT*, *ydbD*, *ydfO*, *ydjJ*, *yerD*, *yetO*, *yfhD*, *yfhF*, *yfhK*, *yfhO*, *yfkM*, *yflA*, *yflD*, *yflT*, *yhcM*, *yhdF*, *yhdN*, *yhxD*, *yjgB*, *yjgC*, *yjgD*, *yjlB*, *yjzE*, *ykgA*, *ykzN*, *ylxP*, *ymaE*, *yocB*, *yocK*, *yojJ*, *yorA*, *yoxB*, *yoxC*, *ypuD*, *yqbM*, *yqgC*, *yqhB*, *yqhY*, *yqjF*, *yqxL*, *yrhK*, *yrvD*, *ysnF*, *ytaB*, *ytxG*, *ytxH*, *ytxJ*, *yugU*, *yuzH*, *yvaA*, *yvaG*, *yvaK*, *yvbG*, *yvgN*, *yvgO*, *yvrE*, *yvyD*, *yvyI*, *ywiE*, *ywjC*, *ywkB*, *ywlB*, *ywmE*, *ywmF*, *ywsB*, *ywtG*, *ywzA*, *yxaB*, *yxbG*, *yxkA*, *yxkO*, *yxnA*, *yxxB*, *yxzF*, *yybO*, *yycD*, *yyzG*, *yyzH*; the full list of the genes and their operons is given in tabular form in the Appendix A). Their functional analysis unsurprisingly showed that they mostly code stress proteins (72%, *n* = 106), and the second largest group belonged to a category “membrane proteins” with 27 genes (18%). The remaining genes of the reported SigB regulon may be expressed under different conditions, require additional factors, or are controlled by different sigma factors or cofactors (the alternatives to SigB control for specific genes are shown in Appendix A). Some of these genes may not be true targets of SigB regulation.

## 4. Conclusions

In this study, using a combined approach of static and dynamic information analyses, we defined the SigB regulon in *B. subtilis* that is active during spore germination and outgrowth. The used approach combined a meta-analysis of literature data with the information about the kinetics of gene expression and promoter sequence analysis. The analysis showed that out of the 411 genes of the theoretical SigB regulon, 146 (35%) were expressed and controlled by SigB during normal growth conditions; most of them coded for stress and/or membrane proteins. The remaining genes of the reported SigB regulon may be expressed under different conditions, are controlled by different sigma factors or in combination with them (mainly SigA), or they require additional factor or cofactors for their expression. The analysis also showed the importance of the organization of the promoter binding sequence, especially the spacing of −35 and −10 elements for promoter recognition by the RNAP–SigB holoenzyme. Consistently, experimentally confirmed class III promoters, which have not been assigned to SigB-dependent regulation by the other analyses presented here (although they were identified as SigB-dependent in the literature), failed to be recognized by the RNAP–SigB holoenzyme in our study. The presented approach shows that in order to identify the regulon active during a specific biological process or specific conditions, the static information (e.g., binding experiments) is not sufficient and other additional source of information is necessary to employ. It also shows that the vast amount of data accumulated in literature and databases can be effectively used to discover new relations even in already well studied systems. The presented approach is general enough to be applied to other systems for which sufficient amounts of data are available.

## Figures and Tables

**Figure 1 microorganisms-09-00112-f001:**
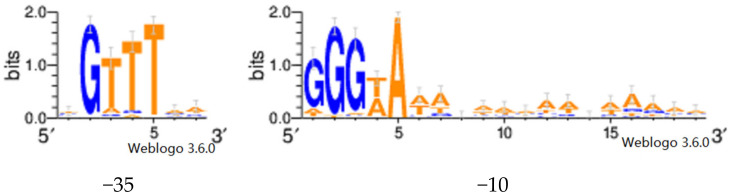
Binding motifs found in genes of SigB regulon. The motifs are ordered from −35 to −10 with spacers of 5–20 nucleotides.

**Figure 2 microorganisms-09-00112-f002:**
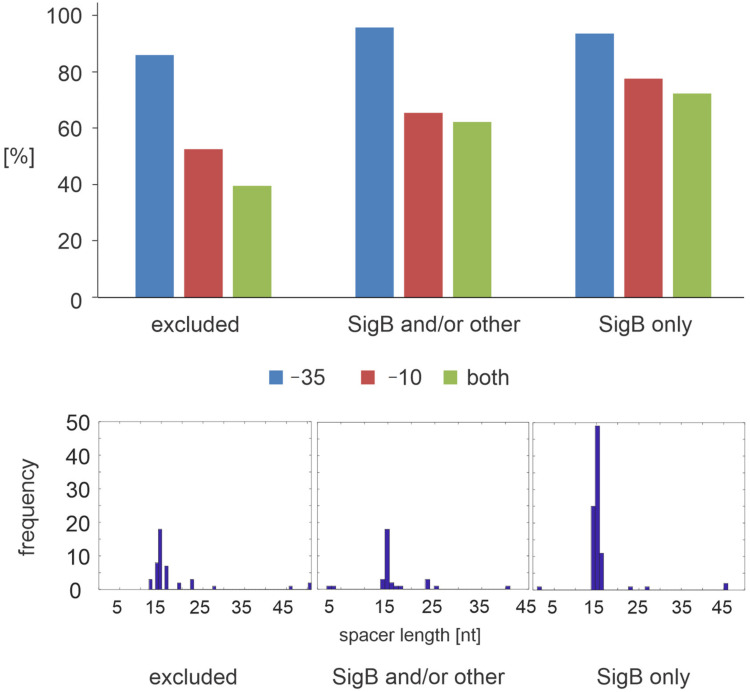
Relative occurrence of −35 and −10 binding motifs and the distance between them in the different SigB regulatory groups. Excluded (type 3)—regulator not found or the genes excluded during preprocessing; SigB and/or other (type 2)—genes for which besides SigB, also another regulator was found; SigB only (type 1)—genes for which SigB was found as the only regulator. The figure shows that the best characteristics were found for type 1 genes where the modeling results were consistent with the binding motifs analysis.

**Figure 3 microorganisms-09-00112-f003:**
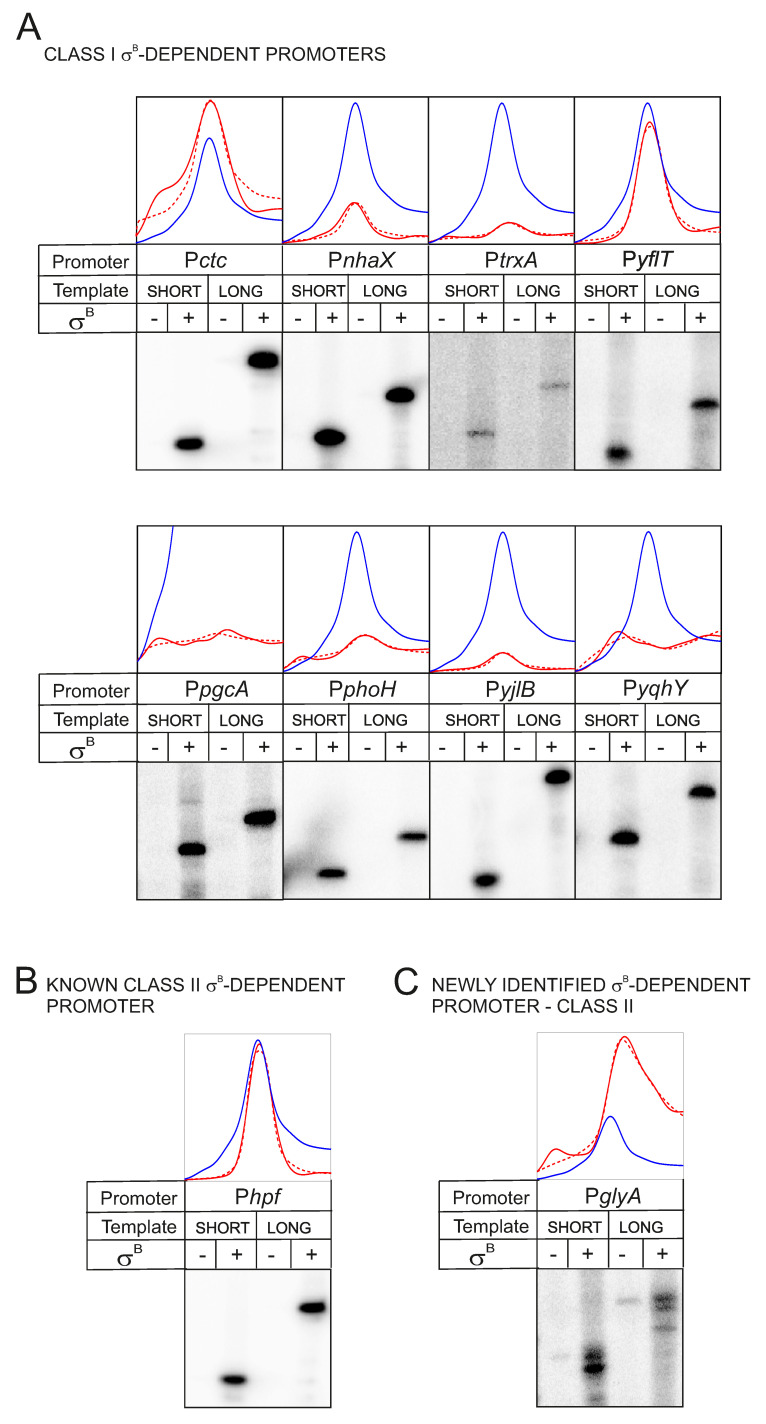
Experimental verification of SigB dependence of selected promoters. The validation was done by in vitro transcription with *B. subtilis* RNAP complexed with SigB (σB). (**A**)–Class I promoters [contain canonically spaced (by 13–15 bp) −35 and −10 elements]. (**B**)–A known Class II promoter contains only the −10 element, highly resembling the consensus sequence. (**C**)–genes of the Class II promoter. Curves represent modeled (red dashed line) and experimental (red solid line) expression profiles of the given gene; blue line is the expression profile of SigB. SHORT and LONG refer to template length (two sizes for each promoter region) to distinguish the orientation of the promoter within the template. −/+ indicate the absence/presence of SigB. In the absence of SigB, only the RNAP core was used. *trxA* was used as a standard.

**Table 1 microorganisms-09-00112-t001:** Experimentally verified genes. Motifs −35 and −10, refer to the binding motifs as defined in Section 2.2.

Gene		CLASS	Transcription	Verified TNX	Spacer Length	Motifs	−35	Spacer	−10
*pgcA (yhxB,gtaC,gtaE)*	BSU09310	I	1	1	16	−35, −10	CGTTTA	TTTTTTGATATCAATT	**GGGTAA**GAACATATAAAGA
*yjlB*	BSU12270	I	1	1	15	−35, −10	TGTTTG	GCGAACCGCTATATG	**TGGAAG**ACAAAAAAGGGAG
*nhaX (yheK)*	BSU09690	I	1	1	15	−35, −10	AGGTTA	ATTGTGCTCAAATTC	**GGGTAG**TAGTGTTGTAAGA
*cadA (yvgW)*	BSU33490		0		15	−35, −10	TGTTTT	TCATTGACACTTTCT	**TGGAAA**ACAACATATAATA
*yqhY*	BSU24330	I	1	1	16	−35, −10	GGTTTC	GCTTGCTAATGAAATT	**GGGTAT**CCTGTAATTATAA
*yflT*	BSU07550	I	1	1	14	−35, −10	TGTTTC	AGGTACAGACGATC	**GGGTAT**GAAAGAAATATAG
*rsgA (cpgA, yloQ)*	BSU15780		0		14	−35, −10	AGATTG	AACCAGGCCAAAAA	**GGGTAC**TATCAAGTAATGG
*ctc*	BSU00520	I	1	1	15	−35, −10	GGTTTA	AATCCTTATCGTTAT	**GGGTAT**TGTTTGTAATAGG
*phoH (yqfE)*	BSU25340	I	1	1	15	−35, −10	AGTTCA	AGAAGGCATTAAATT	**GGGTAA**ACAGGATGTAGAG
*hpf (yviI,yvyD)*	BSU35310	I	1		15	−35, −10	TGTTTC	AGCAGGAATTGTAAA	**GGGTAA**AAGAGAAATAGAT
*glyA (glyC,ipc-34d)*	BSU36900	II	1		-	−10	-	-	**TGGTAA**AAACAAAGAACAG
*yhfP*	BSU10320	II	0		-	−10	-	-	**AGGAAG**AAATAAGATGAAC
*xynA*	BSU18840	III	0		28	−35, −10	TGTTTT	AAATGTATACGAGTGCTACCTCAAAGTC	**GGAAAA**AATATTATAGGAG
*yhfI*	BSU10240	III	0		24	−35, −10	TGTTTA	AAACATGCTTTTTTCAAGAAAAAT	**GGGTAT**ATTGAAGGAGGAC
*pfkA (pfk)*	BSU29190	III	0		35	−35, −10	GGTTTC	ATAGGGAGGATGGAGATCCCTTTTCATTGTTTTTA	**GGGCAA**TGATCATGTTATG
*uppS (yluA)*	BSU16530	III	0		46	−35, −10	TGTTTA	CAGGGGGTTTTTTTGTTAATACTGTTGATTACATTGATTATCAGCA	**GGGAAT**GTAACCTTTTTGG

## Data Availability

Data is contained within the article or supplementary material.

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
