# Peer review of "Kinetic Modeling and Meta-Analysis of the *Bacillus subtilis* SigB Regulon during Spore Germination and Outgrowth"

_microorganisms, 2021, doi:10.3390/microorganisms9010112_

Round 1
Reviewer 1 Report
In the article "Kinetic modeling and meta-analysis of the Bacillus subtilis SigB regulon during spore germination and outgrowth," the authors applied a combined approach using statistical and dynamic data analysis to expand understanding of the role of the SigB regulon in spore germination in B. subtilis. The results of the study made it possible to more accurately determine the structure of the sequence required by SigB for binding and activation of transcription. Using the data obtained, the authors identified a basic set of genes that are specifically expressed during spore germination under the control of SigB.
Author Response
Reviewer does not require making any changes.
Reviewer 2 Report
The manuscript is not cogent. Rather, it is a discursive description of the SigB regulon that functions during spore germination and outgrowth. SigB is a general stress response factor, and, as such, regulates a number of genes. A key question not answered by the authors is what external factors influence the SigB regulon, and gene regulation overall. Identifying "parameters that fit the measured expression profile of the given regulated gene with the SigB profiles as the regulator within the confidence interval" is a reasonable approach. However, the question still remains: "How is the "subset of the SigB regulon" activated? Specifically, what additional transcription factors not indicated in the in vitro assays are required? The author have determined "a", not "the", SigB regulon they believe is involved in spore germination and outgrowth.
Early classical studies of spore germination showed that manganese activates a proteolytic enzyme in spores of B. megaterium which breaks down spore protein and, in turn, stimulates spore germination. Have the authors considered how cations such as manganese and calcium affect the SigB regulon?
Author Response
The manuscript is not cogent. Rather, it is a discursive description of the SigB regulon that functions during spore germination and outgrowth. SigB is a general stress response factor, and, as such, regulates a number of genes. A key question not answered by the authors is what external factors influence the SigB regulon, and gene regulation overall. Identifying "parameters that fit the measured expression profile of the given regulated gene with the SigB profiles as the regulator within the confidence interval" is a reasonable approach. However, the question still remains: "How is the "subset of the SigB regulon" activated? Specifically, what additional transcription factors not indicated in the in vitro assays are required? The author have determined "a", not "the", SigB regulon they believe is involved in spore germination and outgrowth.
Re: The sigB regulon as documented in the literature and the SubtiWiki database involves all the documented regulatory interactions. For some genes SigB was reported as the only regulator, in other cases there are as many as 6 regulators (line 244 and on). How this issue was approached in the manuscript is discussed in paragraph 3.1., particularly in the paragraph starting at line 244. The manuscript already contains what the reviewer requests. Wherever reported, we modeled the influence of all combinations of regulators. The results are summarized in Supplementary Table 2, where the computed regulators are listed. So, the information is in the manuscript. We just did not emphasize it as the paper focuses on the activation of a subset of the SigB regulon that is active during germination and outgrowth, i.e. the situation where no external stress is applied. We believe that such information is important as in the published studies of the SigB regulon, this topic was ignored. We know almost all about what happens under stress, but we do not know what occurs under normal conditions.
Early classical studies of spore germination showed that manganese activates a proteolytic enzyme in spores of B. megaterium which breaks down spore protein and, in turn, stimulates spore germination. Have the authors considered how cations such as manganese and calcium affect the SigB regulon?
Re: No, we have not. Nevertheless, it is important and interesting but it is beyond the scope of this paper.
Reviewer 3 Report
In this manuscript, the authors demonstrate the power of integrating data from archived experimental results, a curated, community-annotated genome database (SubtiWiki), and fresh in vitro gene expression studies to delineate a novel role for a known master regulator of gene expression, in this case the role of the general stress sigma factor of Bacillus subtilis, SigB, in spore germination and outgrowth. In addition, the results also highlight the promoter structural components (-35 sequence, spacer length, -10 sequence) that likely determine whether transcriptional regulation is mediated by SigB only or whether SigB requires accessory factors. I don't feel qualified to comment on the mathematical modeling performed in the analysis. The experimental design as a whole seems well-conceived, however, and the results seem relevant and interesting. I have only a few comments:
(1) Table 2 is not in tabular format; it is a list of comma-separated values and is not very comprehensible to the reader in its present form. Here are two suggestions for possibly improving the presentation:
a) The list would be more compact and comprehensible if genes comprising known operons were grouped and written in condensed format where possible, i.e. ytxGHJ instead of "ytxG, ytxH, ytxJ." Even where condensed format is not possible, grouping genes from the same operon, currently scattered all over the alphabetized list, could be recognized instantly, i.e. ctsR-mcsA-mcsB-clpC-radA.
b) Do the authors think it would be possible and advantageous to produce a two-column table, one column a proposed functional category and the second a list of genes in that category? I think such a list could be thought provoking and perhaps stimulate additional experimentation. For example, the ydaJKLMN operon is involved in synthesis of extracellular polysaccharide, normally present in biofilms. (I find it interesting and somewhat counter-intuitive that these genes would be activated during germination and outgrowth.) I would envision such a table being a highly condensed version of the functional analysis table found in the supplementary materials.
(2) line 42: Since "bacillus" is simply a cell morphology without any phylogenetic meaning, it would be better to use the genus name Bacillus (in italics).
(3) line 92: "Briefly, 168 spores of Bacillus subtilis..." Presumably, the authors intended to say "Briefly, spores of Bacillus subtilis 168..."
Author Response
In this manuscript, the authors demonstrate the power of integrating data from archived experimental results, a curated, community-annotated genome database (SubtiWiki), and fresh in vitro gene expression studies to delineate a novel role for a known master regulator of gene expression, in this case the role of the general stress sigma factor of Bacillus subtilis, SigB, in spore germination and outgrowth. In addition, the results also highlight the promoter structural components (-35 sequence, spacer length, -10 sequence) that likely determine whether transcriptional regulation is mediated by SigB only or whether SigB requires accessory factors. I don't feel qualified to comment on the mathematical modeling performed in the analysis. The experimental design as a whole seems well-conceived, however, and the results seem relevant and interesting. I have only a few comments:
(1) Table 2 is not in tabular format; it is a list of comma-separated values and is not very comprehensible to the reader in its present form. Here are two suggestions for possibly improving the presentation:
- a) The list would be more compact and comprehensible if genes comprising known operons were grouped and written in condensed format where possible, i.e. ytxGHJ instead of "ytxG, ytxH, ytxJ." Even where condensed format is not possible, grouping genes from the same operon, currently scattered all over the alphabetized list, could be recognized instantly, i.e. ctsR-mcsA-mcsB-clpC-radA
- b) Do the authors think it would be possible and advantageous to produce a two-column table, one column a proposed functional category and the second a list of genes in that category? I think such a list could be thought provoking and perhaps stimulate additional experimentation. For example, the ydaJKLMN operon is involved in synthesis of extracellular polysaccharide, normally present in biofilms. (I find it interesting and somewhat counter-intuitive that these genes would be activated during germination and outgrowth.) I would envision such a table being a highly condensed version of the functional analysis table found in the supplementary materials.
Re: We chose to follow suggestion a). We created a new table listing all found genes and their operons in a tabular form as suggested. The table is quite large (115 lines) for printed version so we chose a compromise: we put the mere list of genes from Table 2 as an inline text in parentheses. It starts at line 373 and we moved the comprehensive table to the supplements as the supplementary Table 2. This allows the readers to quickly check the found genes, and the detailed information can be found in supplements.
(2) line 42: Since "bacillus" is simply a cell morphology without any phylogenetic meaning, it would be better to use the genus name Bacillus (in italics).
Re: The suggested change was made.
(3) line 92: "Briefly, 168 spores of Bacillus subtilis..." Presumably, the authors intended to say "Briefly, spores of Bacillus subtilis 168..."
Re: Thanks, the change was made.
Round 2
Reviewer 2 Report
The manuscript remains diffuse. Attention should have been paid to the early classical papers on spore germination and outgrowth. Disregarding some of the important history on this subject is close-minded.